precision medicine; personalized care; patient care planning; quality of life; multimorbidity; learning health systems

**Author for correspondence:**
Arlene S. Bierman,
Email: arlene.bierman@ahrq.hhs.gov

# From precision medicine to precision care: Choosing and using precision medicine in the context of multimorbidity

Arlene S. Bierman ⓘ, Bridget T. Burke, Leeann N. Comfort, Maya Gerstein, Nora M. Mueller and Craig A. Umscheid

Center for Evidence and Practice Improvement, Agency for Healthcare Research and Quality, Rockville, MD, USA

## Abstract

Rapid advances in precision medicine promise dramatic reductions in morbidity and mortality for a growing array of conditions. To realize the benefits of precision medicine and minimize harm, it is necessary to address real-world challenges encountered in translating this research into practice. Foremost among these is how to choose and use precision medicine modalities in real-world practice by addressing issues related to caring for the sizable proportion of people living with multimorbidity. Precision medicine needs to be delivered in the broader context of *precision care* to account for factors that influence outcomes for specific therapeutics. Precision care integrates a person-centered approach with precision medicine to inform decision making and care planning by taking multimorbidity, functional status, values, goals, preferences, social and societal context into account. Designing dissemination and implementation of precision medicine around precision care would improve person-centered quality and outcomes of care, target interventions to those most likely to benefit thereby improving access to new therapeutics, minimize the risk of withdrawal from the market from unanticipated harms of therapy, and advance health equity by tailoring interventions and care to meet the needs of diverse individuals and populations. Precision medicine delivered in the context of precision care would foster respectful care aligned with preferences, values, and goals, engendering trust, and providing needed information to make informed decisions. Accelerating adoption requires attention to the full continuum of translational research: developing new approaches, demonstrating their usefulness, disseminating and implementing findings, while engaging patients throughout the process. This encompasses basic science, preclinical and clinical research and implementation into practice, ultimately improving health. This article examines challenges to the adoption of precision medicine in the context of multimorbidity. Although the potential of precision medicine is enormous, proactive efforts are needed to avoid unintended consequences and foster its equitable and effective adoption.

## Impact statement

Precision care encompasses a person-centered approach to precision medicine to inform decision making and care planning by taking multimorbidity, functional status, values, goals, preferences, social and societal context into account. This article examines challenges that need to be addressed to realize the promise of precision medicine in the context of multimorbidity. It also discusses how partnerships across the research enterprise; between researchers, practice, and policymakers; with patients, caregivers and communities; between clinical medicine and public health; and across sectors could advance effective real-world adoption of precision medicine.

## Introduction

Rapid advances in precision medicine hold the promise of dramatic reductions in morbidity and mortality for a growing array of conditions. However, to realize the potential benefits of precision medicine and minimize possible harms, real-world challenges for translating this research into practice will need to be addressed. Foremost among these is how to choose and use precision medicine modalities in real-world practice by addressing the many issues related to providing care to the large proportion of the population living with multimorbidity.

Multimorbidity, the presence of multiple coexisting physical and/or mental health conditions, functional limitations, and social risks is exceedingly common in clinical practice worldwide but uncommon in clinical trials (Hanlon et al., 2019). Application of novel precision medicine therapeutics in the context of multimorbidity necessitates consideration of underlying

conditions, functional status, and a person's goals of care (Tinetti et al., 2016; Blaum et al., 2018). Socioeconomically disadvantaged populations have a higher prevalence and burden of multimorbidity and are also underrepresented in clinical trials. Scale and spread of precision medicine will require consideration of, evidence for, and strategies to inform decision making about the use of therapeutic advancements, while also addressing the challenges inherent in determining their effectiveness and delivering these interventions among people living with multimorbidity in diverse populations.

The balance of risk and benefits of a given treatment will differ in the context of underlying conditions. For example, treatments with cardiac or pulmonary toxicity will require different considerations for use among individuals with heart failure or chronic obstructive pulmonary disease, respectively. There may be drug–drug and/or drug–disease interactions that need to be considered in planning care. Furthermore, issues of access to care, costs, and social supports also need to be considered. It is only by attention to these issues that it will be possible to achieve the ultimate objective of optimizing individual and population health.

Precision medicine commonly refers to treatments tailored to individuals based on genomics and other biomedical advances in order to apply treatments to those most likely to benefit, though it may include other factors such as environment and lifestyle (National Research Council, 2011). Delivering precision medicine in the broader context of *precision care* will enable addressing the multiple factors that may influence outcomes for a given treatment. Precision care encompasses a person-centered approach to precision medicine informing decision making and care planning by taking multimorbidity, functional status, values, goals, preferences, social and societal context into account (Bierman and Tinetti, 2016). Precision care entails developing individualized care plans to help people maximize their functional status and well-being and to achieve their goals.

The concept of tailoring care to individual goals and preferences is well established, and a central tenet in the care of older adults and people living with multimorbidity (Tinetti et al., 2019; Bierman et al., 2021). One-size-fits-all care has always been particularly problematic for people with multimorbidity for whom interactions between multiple physical and/or mental health conditions and treatments create unique care needs. People living with multimorbidity face nuanced and often difficult care decisions as they experience competing demands of symptoms, treatments, self-care, in the context of available social and financial resources. As a result, traditional single disease, evidence-based outcomes may not be their highest priority or they may prioritize evidence-based, disease-specific outcomes from one condition over others. Therefore, engaging in discussion about which outcomes that matter most is crucial. Burden of treatment is often a major concern (Spencer-Bonilla et al., 2017). Designing plans of care to minimize treatment burden can benefit quality of life, address competing demands, and alleviate financial pressures (Leppin et al., 2015).

Designing dissemination and implementation of precision medicine around the broader concept of precision care would improve person-centered quality and outcomes of care, target interventions to those most likely to benefit thereby improving access to new therapeutics, minimize the risk of withdrawal from the market from unanticipated harms of therapy, and advance health equity by tailoring interventions and care to meet the needs of diverse individuals and populations. This paper examines challenges that need to be addressed to advance the adoption of precision medicine and realize its promise in the context of multimorbidity.

## Continuum of translational research and multimorbidity

There is a long road from scientific discovery to routine application in clinical practice and societal benefit. The uptake of therapeutic advancements often takes decades. To foster and accelerate their adoption, the full continuum of translational research needs to be addressed: developing new approaches, demonstrating their usefulness, and disseminating and implementing their findings, engaging patients throughout the process (Figure 1). This encompasses basic science, preclinical and clinical research, followed by implementation into practice and ultimately impact on public health. Table 1 illustrates how multimorbidity can be considered in each step of the process from animal models, to phase 1 studies, clinical trials, effectiveness studies in real-world practice, and assessments of population health impact.

## Evidence to inform precision care

When considering evidence to inform precision care, the primary focus is on rigorously controlled randomized trials examining the efficacy of precision therapeutics targeting specific genotypes. The evidence needed to inform precision care is broader and richer than effectiveness studies evaluating new diagnostic and therapeutic modalities (Abou-el-Enein et al., 2021). Randomized clinical trials (RCTs) typically assess interventions in highly selected and controlled populations with few comorbidities other than the conditions of interest. RCTs report outcomes from the enrolled populations, and at times from pre-specified subgroups. Increased attention is being directed at the need to consider the heterogeneity of treatment effects and the need to understand factors associated with intervention effectiveness in each individual (Kent et al., 2018). Evidence is also required to understand the impact of interventions in real-world settings, where these interventions are administered to those living with multiple chronic conditions. Furthermore, social factors that affect access, decision making, and outcomes need to be considered. Assessing evidence in clinically and socioeconomically diverse populations would help us to better understand the long-term benefits and harms and potential trade-offs of interventions in patient populations commonly encountered in the real world, for example, those with diabetes, heart failure and/or depression or a combination of these conditions. Study designs that can provide the needed evidence include pragmatic trials and registry studies. Adoption of a core set of harmonized measures will allow for pooling of data and metanalysis to accelerate evidence generation (Beckmann and Lew, 2016; Leavy et al., 2019).

In addition to clinical evidence of effectiveness, additional evidence is needed to inform decision making about when, for whom, and how to use diagnostic and therapeutic advancements. Evidence examining the varied preferences of patients and clinicians can help inform shared decision making (SDM) to facilitate precision care (Umscheid, 2009), particularly when these preferences reflect real-world concerns, such as polypharmacy and financial toxicity of medical care (Schnipper et al., 2015). Evidence examining not only the "what" but also the "how" is also fundamental to informing precision care. Such evidence addresses those challenges frequently encountered by health systems, clinicians, and patients – not "what" to do, but "how" to provide care that makes it easy for patients to get what they need when they need it every time, across

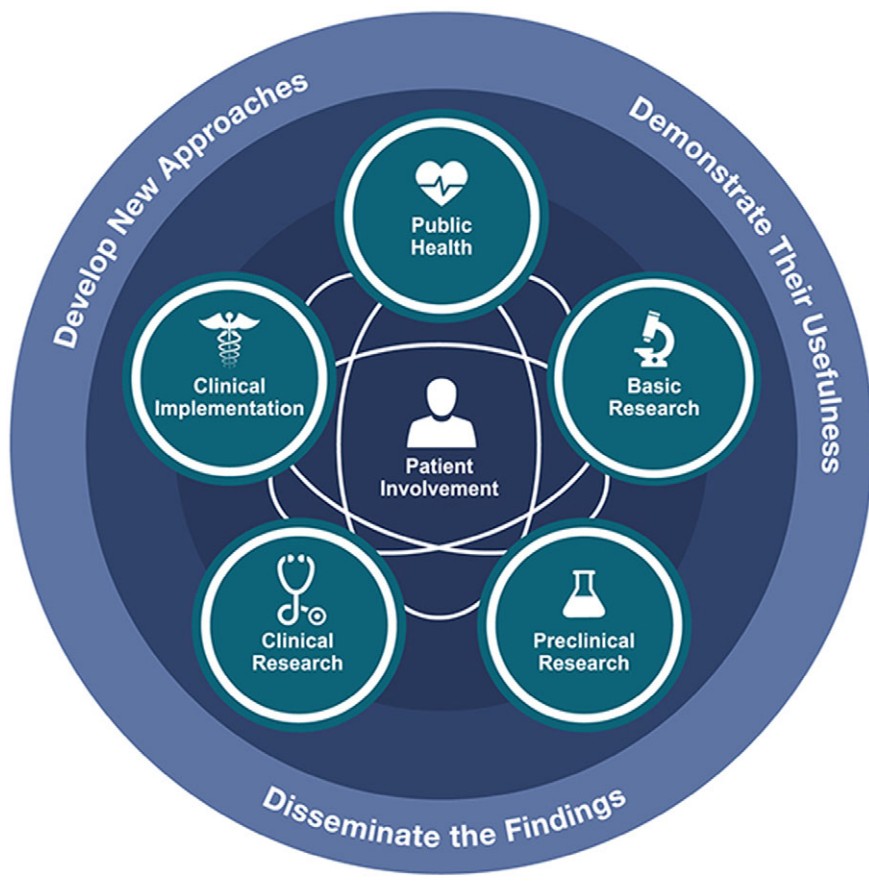

Credit: National Center for Advancing Translational Sciences

**Figure 1.** T0–T4 continuum of translational research to improve individual and population health.

**Table 1.** T0–T4 research in the context of multimorbidity

| | |
|---|---|
| T0: Basic science and pre-clinical research | Consideration of multimorbidity when developing or selecting animal models |
| T1: From bench to bedside | Consideration of multimorbidity in early development and testing of diagnostics and therapeutics |
| T2: From bedside to clinical practice | Recruit individuals with multimorbidity in clinical trials to assess efficacy, effectiveness, and safety in presence of co-existing illness, physical limitations, and social context |
| T3: From clinical practice to widespread care delivery | Study the effectiveness of the intervention in real-world practice among individuals with multimorbidity as well as the effectiveness of implementation strategies in diverse clinical settings and populations |
| T4: From health care delivery to impact on the community, public health, and public policy | Determine impact on health systems, population health, and implications for public policy including issues related to multimorbidity, cost–benefit analysis, and input from diverse populations and communities |

*Note*: T0–T4 adapted from Tufts Clinical and Translational Science Institute (https://www.tuftsctsi.org/about-us/what-is-translational-science/)

their lifespan. Traditional recommendations from clinical practice guidelines are not yet able to address the precision care needs of diverse patient populations, but by leveraging big data available for each patient on their clinical and social determinants of health, it is conceivable that one day the rich and diverse sources of evidence described above could inform individualized recommendations, that would enable more precise care and SDM (Eddy et al., 2011).

An evidence base of harmonized outcome measures from representative patient samples, together with accounting for aggregate financial and non-financial costs, will allow governments and payers to calculate value and cost-effectiveness of therapeutic innovations from a societal perspective. These macro-level evaluations can both identify opportunities to adapt precision care implementation strategies and enable comparisons of the social impact of precision care investments to other opportunities.

### Comprehensive care planning

With advances in computing, artificial intelligence, and patient-generated and high-throughput data, it is important to ensure that patients' goals, preferences, and social context are not lost among

the increasing volume of information. To ensure that patients remain at the center of care, all members of the care team, including patients and their caregivers, will need direct access to a comprehensive longitudinal shared care plan that documents all health conditions, medications, goals, preferences, and social context. The process of care planning requires collaboration among patients and clinicians to proactively discuss and document information critical to providing precision care, including roles, strategies for supporting and empowering patients, plans for engaging in SDM, and plans to balance evidence-based care with patient preferences and treatment burden (Burt et al., 2014). Comprehensive care plans enable health care clinicians to focus on proactive, prevention-oriented care, make clinical decisions in the context of the patient's specific needs, and facilitate communication and coordination among all members of the care team (Holland and Lee, 2019). It is important that care planning incorporates processes for updating plans as new health care options become available, along with new information available from continuous monitoring and tracking of progress on goals. Effective care plans help tailor care to patient goals and preferences during acute, episodic care, as well as during health maintenance and care transitions, helping both patients and clinicians better manage multimorbidity and prevent additional impairments (Baker et al., 2016).

The need to incorporate many different types and sources of data requires mechanisms to share care plans with all members of the clinical team (including primary care, specialty care, behavioral health, pharmacy, community health workers, and many other relevant team members within and across organizations), as well as patients and their caregivers, to ensure that the right information gets to the right person at the right time. Because care plans themselves can cause harm if a patient ends up with multiple competing and sometimes conflicting plans, the use of interoperable comprehensive shared electronic (e-) care plans (CSeCP), electronic tools that use health information technology to document and share information, can provide all members of the care team with electronic access to information critical to their role (Consumer Partnership for eHealth (CPeH), 2013). The use of interoperable shared electronic care plans is an active area of development, particularly as standards and clinical terminologies become harmonized. For example, the Fast Healthcare Interoperability Resources (FHIR) specification, a standard for exchanging healthcare information electronically, provides a mechanism to exchange the rapidly growing volume of health data in a lightweight, real time, and secure manner, regardless of where the health data are stored (Office of the National Coordinator for Health Information Technology, 2019). Further development of infrastructure and technologies to support data sharing and visualization, along with changes in clinical practice, medical education, and ensuring access in rural and socioeconomically disadvantaged populations will be required to realize the promise of e-care plans in precision care (Backonja et al., 2018; Norton et al., 2022). Availability of comprehensive person-centered data from e-care plans can also serve as a rich source of data to study the effectiveness of precision medicine therapeutics in people with multimorbidity.

Clinical decision support can provide clinical teams and patients with knowledge and person-specific information, intelligently filtered or presented at appropriate times, to inform care planning. These digital tools offer clinicians and patients a means to access, understand, and apply personalized data and scientific evidence. Further development, including new advances in artificial intelligence to integrate medical, genetic, patient-reported, and community data can enhance the functionality and utility of clinical decision support (Romero-Brufau et al., 2020). This will aid clinicians in communicating increasingly complicated personalized risk information while facilitating collaborative discussions on the treatment and management of multiple conditions (Zipkin et al., 2014). CDS tools with integrated AI that successfully address the varying, sometimes competing needs of patients with multimorbidity will need to be co-developed with patients, healthcare professionals, health systems, developers and other system level stakeholders as equal partners (Mistry, 2019; Silcox et al., 2020).

## Shared decision making and precision care

SDM is a collaborative process in which patients, their families, and caregivers work together in partnership with members of their care team to make health care decisions. In SDM, healthcare decisions are informed both by the care team's medical and scientific knowledge and understanding of available interventions and the patient's individual goals, preferences, and circumstances (Hargraves et al., 2019). SDM is essential to the delivery of quality, patient-centered care for persons with multimorbidity who have many opportunities to make and revisit complex medical decisions and are often best positioned to evaluate the tradeoffs in benefits and burden within the context of their lives (Tinetti et al., 2008; Muth et al., 2014; Chi et al., 2017).

Rapid advances in genetic and molecular testing and parallel advances in the application of patient and community reported measures stand to both enhance and complicate SDM for persons with multimorbidity (Bradley et al., 2016; Krist et al., 2016). SDM should also play an important role in helping patients navigate the expanding landscape of genetic and molecular testing and make decisions about whether to pursue the testing and screening options available to them. There are several features of genetic and molecular testing that position SDM as both a complicated and essential component of consultative care. Genetic or molecular testing can have wide-ranging implications for the patient and their family. It can, for example, affect family dynamics by indirectly informing other relatives of their risk. Genetic and molecular test results may also be perceived by patients as certain when many of these tests can only infer increased risk. Treatments may also not always be available to address the results of predictive genetic or molecular tests. Thus, when patients undergo genetic testing, they may be entering a complicated process where risk probabilities and uncertainty need to be balanced alongside patient goals and preferences.

Furthermore, there are increasing opportunities to combine data sources on social determinants, patient preferences, and genomic and biological markers so patients, in collaboration with their families and clinicians, can make more informed, individualized decisions about their healthcare. These advances will also pose new challenges to the implementation of SDM. To achieve SDM, clinicians and patients need time, skills in facilitated communication, and easy access to clinically useful knowledge (Fraenkel and McGraw, 2007; Joosten et al., 2008; O'Connor et al., 2009). Members of the care team including primary care clinicians will need ready access to information on the benefits and harms of new diagnostic and therapeutic modalities to guide these discussions. This capacity, which currently does not exist in most practices, will become increasingly important as the volume of available and actionable personal health information grows and the landscape of molecular and genetic testing options for patients with multimorbidity increases in size and complexity. For SDM to become a routine component of practice in the age of precision care,

workflows will need to be adjusted for more efficient division of responsibility across care teams and digital health solutions, such as by using evidence-based artificial intelligence and clinical decision support tools to access, intelligently filter, understand, and apply personal data alongside scientific evidence.

## Health systems and models of care

Health systems play an essential role in developing and implementing models of person-centered precision care. By building infrastructure and aligning incentives across clinicians and payers, learning health systems can create models of care that provide clinicians with the time and access to clinical knowledge needed for precision care processes including comprehensive care planning, addressing social risks, heath equity, and SDM (Mas et al., 2021; Easterling et al., 2022). Clinicians can meet the challenges resulting from quickly evolving clinical evidence and the complexities of caring for those with multimorbidity if information technology that provides access to updated evidence is available and easily usable (Guise et al., 2018), through tools such as clinical pathways, decision aids, and other clinical decision support systems (Borsky et al., 2019; Flores et al., 2019; Bartlett et al., 2022; Easterling et al., 2022).

Learning health systems are uniquely suited to integrate the two primary functions of health care – *caring* and *learning* (Montori et al., 2019). They can play a particularly important role as they undertake the cycle of evidence synthesis, implementation, and generation. The iterative processes of designing, implementing, and evaluating clinical interventions and models of care can provide the evidence needed for widespread dissemination and implementation. Learning health systems have the opportunity to use their robust data collection and analysis infrastructures to generate new evidence on the effectiveness of diagnostic and therapeutic advancements in real world settings as well as effective ways to organize and deliver care (Easterling et al., 2022). Continuous evidence generation by using qualitative and quantitative data and analytics will help determine both the effectiveness of clinical interventions in the context of co-existing illness and social factors, and strategies for care delivery and quality improvement.

Given their access to data on clinically and socioeconomically diverse populations and a broad spectrum of affiliated clinical specialties, systems are well-positioned to design and evaluate models of care that integrate evidence related to the treatment of multiple chronic conditions. Many learning health systems cultivate ties to organizations and resources within their broader communities, enabling them to provide care that incorporates precision medicine and addresses the unique constellation of needs that extend beyond clinical management by integrating social and community services and supports. As health systems cultivate and disseminate innovative models of precision care, they are creating the standards and expectations for how patients will shape their care in an era of data-driven medicine. The norms established in these early demonstrations – including costs and measures of success – will have a lasting influence on how precision care is delivered across specialties, settings, and geographies.

## Social context and precision care

Equitable, patient-centered care includes the active participation of patients in goal setting, ensuring services necessary to address needs are available and accessible, and supporting the development self-

management skills. Efforts to achieve this include the tailoring of care plans to ensure the integration of patient preferences, and enhanced community partnerships to coordinate access to care or the provision of services to address social risks. Precision care lives in this framework: it should always be delivered in a human and social context and shaped by the social and behavioral determinants of health, as well as patient preferences, alongside genomic and other biologic information. Integrating precision medicine into frameworks that address social needs is integral to precision care.

Dynamic contextual factors must be assessed research (Bayliss et al., 2014). Furthermore, clinical research should be conducted with an eye toward implementation, integrated with practice in an iterative loop to ensure real-world relevance while informing existing implementation science models (Chambers et al., 2016). This can be achieved through identifying information and service needs (e.g., community resources to address social needs, leveraging information on payer coverage to avoid financial strain and toxicity); iteratively capturing data, especially patient and provider-reported outcomes, and using those data to inform ongoing clinical and community practice. Furthermore, there is a need for actionable tools and resources to facilitate addressing the social determinants of health in decision making and care planning (Glasgow et al., 2018). Without a shared decision-making process that accounts for individuals' specific socioeconomic context, the added costs of precision medicine technologies may exacerbate existing strain. Information on patients' available resources and supports should inform access and payment models to foster equitable access.

## Discussion

We are at the cusp of a revolution in medicine on a par with the discovery of antibiotics for the treatment of infectious disease. Although the potential of precision medicine is enormous, proactive efforts are needed to optimize benefits, minimize harms, avoid unintended consequences, and to foster equitable and effective adoption of advances in diagnostics and therapeutics. As diagnoses and treatments move from phenotype to genotype, we risk perpetuating a single disease model in which treatments become more precise and conditions become more precisely treated, but we continue to neglect the interactions between multiple conditions and multiple treatments, as well as interactions between physical, social, and emotional determinants of health. Precision medicine modalities will be subject to the same quality challenges as all other clinical interventions — underuse, overuse, and misuse.

Precision medicine delivered in the context of precision care would foster caring for people, their families, and communities in a way that is aligned with their preferences, values, and goals, in a respectful manner that engenders trust and provides them with understandable information needed to make informed decisions. The full continuum of translational science needs to be embraced and supported including foci on person-centered care, functional health outcomes, and population health. The implications for the funding, design, and conduct of research are substantive at the patient, clinician, practice, health system, and societal level.

Evidence is the linchpin of constructive adoption of precision medicine. This includes evidence on the efficacy and effectiveness of diagnostic and therapeutic modalities in individuals with different constellations of multimorbidity and in diverse settings and populations. With the rapid pace of discovery, we will need investment in living systematic reviews and guidelines brought to the

point of care with living clinical decision support. Advancements in the use of real-world data can provide required evidence on how to choose and use these interventions in real-world practice. These data can provide the capacity for post-market surveillance to identify harms that might manifest when new therapeutics are used more widely as well as long-term outcomes not assessed in smaller, shorter term clinical trials. Health services research and implementation science can provide the evidence for strategies to support wider uptake and assure access, quality, and safety in their use.

Routine adoption of precision care will require significant changes in the culture, organization, and delivery of health care. Patients, clinicians, and communities in partnership can work together to redesign care to facilitate comprehensive care planning and SDM informed by tools and resources that provide access to the best available evidence tailored to their unique needs. Practices and health systems can provide the infrastructure to make this possible, generate the evidence to continually improve quality and outcomes of care, and build partnerships with social and community services, public health, and non-medical sectors such as housing and transportation to help address social risks. This transformation will require the support of governments and payers to align payments and incentives with desired outcomes.

From a societal perspective a number of thorny questions will need to be answered. Rising costs threaten health system sustainability, and the high costs of precision medicine will put increasing strain on these systems. How will coverage decisions be made? What evidence, including cost-effectiveness analyses, will be needed? What constitutes value? How will pricing be determined? How will provisions be made to ensure equitable access, use, and outcomes? How should resource allocation be done? How should opportunity costs such as investments in public health or interventions to address the social determinants of health be considered?

The U.S. Agency for Healthcare Research and Quality (AHRQ) is engaged in efforts that support evidence synthesis; improvement in care delivery by learning health systems and primary care practices; use of digital health applications and clinical decision support to improve care; implementation of evidence into practice; patient engagement to learn identify goals of care and priorities to enable the co-creation of care plans, as well as data and analytics that inform all these efforts. Stronger linkages between basic science, clinical research, health services research and implementation science could help us to develop a coordinated approach to better choose and use precision medicine in the context of multimorbidity.

In summary, a proactive approach to addressing the multiple challenges inherent in delivering precision care in the context of multimorbidity could accelerate progress. Dialogue, engagement, and partnership across the research enterprise; between researchers, practice, and policymakers; with patients, caregivers and communities; between clinical medicine and public health; and across sectors to address the social determinants of health can all contribute to realizing the promise of precision medicine. That's a tall order and aspirational but there can be enormous benefit in moving in this direction.

**Open peer review.** To view the open peer review materials for this article, please visit http://doi.org/10.1017/pcm.2023.8.

**Disclaimer.** The findings and conclusions in this document are those of the authors, who are responsible for its content, and do not necessarily represent the views of the Agency for Healthcare Research and Quality (AHRQ). No statement in this review should be construed as an official position of AHRQ, or the U.S. Department of Health and Human Services.

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
