## [Reviewer Report]

*Comments to Author*: Thank you for asking me to review this thoughtful and informed opinion piece about an important area. I enjoyed the analysis of the issues and the vision of what it would take to move in the direction of ‘precision care’. (Is it necessary to create a new term ‘precision care’? This seems to be simply another name for person-centred care, as I understand it in its fullest sense? If it is, could we stick to person-centred to avoid proliferation of terms and consequent confusion. If it isn’t, please explain the difference).

My main comment is that the article is, as the authors say in their final sentence on P28, aspirational and also rather abstract. Although the authors have tried to give some examples and indications of concrete actions, I wonder if this could go further and include some recommended specific actions. I also felt that the article hovered somewhere between being US system specific and an attempt to make it more generic or generalisable. Perhaps it would be better to be explicit that the analysis and recommendations are based on the US health system. Alternatively, the references to payers and coverage could be made more generic to cover all systems of healthcare funding.

The whole reads well and the conclusion draws it nicely together. I recommend writing the abstract afresh though, so that it does not simply re-produce various extracts from the paper but summarises what the paper is about and its essential argument. I think the abstract should also include the aim of the article.

Minor comments/typos

P18 line 204 could include something about patients’ own understanding and experience of their conditions, which is particularly relevant in consulting with patients with long-term conditions.

P20 line 246-7 - this is an example of where more specific suggestions and detail would be useful

P21 line 247 – also please define a learning health system

P26 line 351 – typo ‘to’ missing

P27 line 365 - typo ‘learn’ or ‘engage’?

---

## [Reviewer Report]

*Comments to Author*: Thank you for the opportunity to review this Review article on Precision Medicine for Multimorbidity. Overall I found there was a lot in this manuscript that I agreed with. If this was a conference plenary I would be one of those people in the audience nodding along. Equally, as a patient advocate and a caregiver, I’m deeply aware of how far “what could be” is from “what actually is”. We are in a trying time for medicine at the moment, and so I’ve tried to approach this from the perspective of a critical reader wary of platitudes rather than an enthusiastic member of the congregation. Those are the people who most need convincing after all.

General comments:

1.) As a reader coming to the concept of “Precision Care” for the first time, I feel this piece runs the risk of being nebulous in its definitions. Personally, I had thought that care that is “aligned with preferences, values, and goals, in a respectful manner that engenders trust and provides understandable information needed to make informed decisions” – is not *precision* care, it’s just, well, care. To state it another way, what’s the opposite of precision care? Is it imprecise care? Or bad care? Because if we’re making the case for good care being better than bad care, well, nobody’s going to disagree with that, so it’s not clear what the thrust of the argument to be made is, at least in my initial look at the abstract.

2.) I had thought one of the key vulnerabilities for Precision Medicine goes that the more “precise” (i.e. smaller) your own particular point in the Venn diagram (e.g. you’ve got a rare genetic mutation in a rare disease or you’ve got an unusual arrangement of comorbidities), the less evidence there is and therefore the wider our confidence intervals must be. In other words, picking a smaller and smaller pool of evidence to draw upon, even if it does reflect a “patient like me” is doing so from a much smaller evidence base with greater risks of bias and inaccuracy. Therefore, someone being treated with a Precision Medicine approach might be getting poorer treatment from an evidence-based medicine perspective.

Specific comments:

1.) Introduction – This section (and throughout) is relatively lightly referenced for a Review. There is much to agree with here (e.g. under-represented populations in trials, issues with multimorbidity, one-size-fits-all not ideal) but I’m not seeing any citations here for interested or critical readers to drill down on what exactly is meant. For instance, please prove to the reader that there is such a thing as “one size fits all care” – where is it? And how do we *know* its been problematic for those with multimorbidity? Point the reader to a systematic review ideally or some other high level evidence. I imagine no clinician sits there and thinks “today I will provide one size fits all care because I’m tired or I’m busy or I can’t be bothered”. So we would need some sort of research to establish that this is occurring, that it’s defined clearly, and that it leads to poorer outcomes – we can’t take it as a given. Here’s a counter-factual – maybe clinicians don’t make *that much* impact on the lives of patients because they work in a flawed health system that is slow to respond, incurs major costs, and only has limited interventions in the context of other bigger socioeconomic factors. Patients don’t hear what their clinician tells them, don’t pick up their meds, and are non-adherent to them. So, whether the clinician is providing precision care or one-size-fits-all care doesn’t have that big an effect size in the grand scheme of things. I’m not saying this is the case, I’m just suggesting that referencing the best evidence to support points like “Burden of treatment is often a major concern” would help readers follow the argument.

2.) Evidence to inform precision care. The authors argue that traditional RCTs focus on controlled populations, but there is a greater need for real world settings. This feels like the expected spectrum of evidence, from on the one hand explanatory trials meant to test falsifiable hypotheses under controlled conditions to on the hand pragmatic trials meant to establish real-world evidence. But these are different things for different purposes – a regulator would not accept a RWE study to assess the safety and efficacy of a new approach, and conversely a payer doesn’t need a new, tightly controlled RCT to set a price or coverage. So what’s the argument here? Is it that we need to shift some resources from explanatory to pragmatic? That we need to increase the resources so we can cover both? Or that we need to intertwine the two approaches through common standards, metrics, approaches? I’m seeing that more evidence is needed, but more evidence is always needed, so I’m not quite sure what the specific call or ask is here.

3.) Comprehensive care planning – There is a bold assertion being made here that “To ensure that patients remain at the center of care, all members of the care team, including patients and their caregivers, will need direct access to a comprehensive longitudinal shared care plan that documents all health conditions, medications, goals, preferences, and social context.” – Again, as someone who has spent decades building digital tools that do that just that, I would like to think that’s true. But there’s not been a case built in *this* paper or in evidence cited to support that claim. Again, the counter-factuals: i.) It’s possible to have patient-centered care without that shared care plan so long as the care is happening anyway or a caregiver is actively involved in management to ensure it’s happening, ii.) maybe only *some* members of the care team have access and patient-centric care is still occurring, iii.) if the care plan does not include goals, preferences, or social context, it’s still possible that patient-centric care is occurring, iv.) it’s possible that a care team spending resources and time documenting things is not spending the time actually delivering the care, and therefore in this approach the patient is receiving poorer outcomes, v.) some patients don’t want this, they want the clinician to take a more paternalistic driving role and the investment in technology and resources will be counter-productive for this subgroup who will feel railroaded into using a shared care record they don’t want.

4.) Line 190 – “Clinical decision support can…” – There’s only a single reference here dating back to 2014 (Zipkin) which is pretty old in terms of digital tools, AI, etc. Rather than hearing how these things *could* help it might be more interesting in a review to hear *how that’s going*. For instance now we know clinicians spend a lot more of their time tapping keys on the EMR. Better claims data for insurance companies and better billing for the hospital, sure, but is the impact of that good for patient interactions? There is always a tradeoff to be had – readers would be keen to hear an expert view from AHRQ of how we’re doing not just the potential.

5.) Based on the abstract and title, I would have expected to read a bit more specifically on the multimorbidity. I might be being unimaginative but a worked example throughout might be useful. For instance say someone’s got diabetes, RA, and depression. How does a precision care approach differ from the one size fits all approach? These are well evidenced and guidelined conditions – but to my knowledge there’s not much of a genetic component to management. The drugs may well interact, the burden of treating each independently is high, and there’s a role for coordination to prevent worsening. If that’s not a good example perhaps pick some other combination – but I had trouble making concrete just what the authors meant without an example – it could be a case study in a box out. I had trouble imaging who was doing all this new work – was it a PCP? Are we talking about speciality management of say cystic fibrosis (which btw I think is a space that this is all being done quite well, for bottom-up reasons not top down). Helping to make this real for the reader would help connect to the message and arguments.

6.) Discussion – The antibiotic comparison might be a little hyperbolic given my earlier comments (and penicillin was one-size-fits all!) and it’s a very different form of technology to the entire paradigm shift of healthcare delivery outlined here.

7.) Discussion – Closing arguments – Line 353 onwards. I don’t disagree that these are all thorny questions, but they will be familiar to readers. If AHRQ is going to be part of the solution I think this would be an excellent platform not to pose rhetorical questions or motherhood + apple pie to do science better, but to state what your investment thesis will be. E.g. “We believe that changing from fee for service to value based care, layered upon rich data supporting patient-centric care, is the only way forward. We’re so committed we’re going to fund infrastructure that nobody else will fund, we’re going to shift our focus to reflective practice solely on these topics, we’re cutting programs that only compare drug A and drug B, and we’re making it a condition of funding that multimorbidity and social determinants is included in every study we commission” – I’m making that up, clearly, but readers don’t come to a review by such esteemed experts merely to see that they, too, see a big raft of uncertainty. Tell us what you believe, tell us where you’re placing your bets. Be bold.

---

## [Reviewer Report]

*Comments to Author*: Thanks to the authors for additions of references and clarification of arguments, my comments have all been adequately addressed.

---

## [Reviewer Report]

*Comments to Author*: The definition of the terms 'precision medicine' and 'precision care' is helpful, as well as explanation of the relationship between them and person-centred care. This improves the accessibility of the article as does the clearer statement of the aim.